# Patient with Aberrant Internal Carotid Artery in the Middle Ear Presenting with Rare Symptoms of Mixed Hearing Loss and Postauricular Pain: A Case Report

**DOI:** 10.3390/medicina58111672

**Published:** 2022-11-18

**Authors:** Bo-Nien Chen

**Affiliations:** 1Department of Otolaryngology-Head and Neck Surgery, Hsinchu MacKay Memorial Hospital, Hsinchu 300, Taiwan; 4622@mmh.org.tw; Tel.: +886-3-611-9595; 2Department of Medicine, Mackay Medical College, New Taipei 252, Taiwan

**Keywords:** aberrant internal carotid artery, mixed hearing loss, postauricular pain, oxcarbazepine

## Abstract

Aberrant internal carotid artery in the middle ear (aICA) is a rare congenital malformation in which the internal carotid artery passes through the tympanic cavity without being separated by bone. A pulsatile vascular mass can be observed in the tympanic cavity of patients with aICA. The diagnosis of aICA may be challenging because improper surgery or treatment can lead to iatrogenic injury, including massive hemorrhage. The aim of this case report was to describe a 39-year-old woman with aICA presenting with mixed hearing loss and postauricular pain. We provided detailed clinical images of the patient to illustrate how aICA can be diagnosed. Because this patient’s aICA had no risk of bleeding, close observation, pharmacological therapy, and regular follow-up were applied. The patient’s postauricular pain was significantly improved after treatment with oxcarbazepine. At the time of this manuscript’s preparation, the patient had been monitored for 10 years and had stable mixed hearing loss with no other complications. Based on the management of this patient, oxcarbazepine can improve aICA-associated postauricular pain, and conservative treatment should be prioritized in cases of aICA without a risk of bleeding. Further studies in a large cohort are required to confirm our findings and recommendations.

## 1. Introduction

Aberrant internal carotid artery in the middle ear (aICA) is a rare congenital malformation in which the internal carotid artery (ICA) passes through the tympanic cavity without being separated by bone. Thus, a vascular mass is seen in the patient’s tympanic cavity. The diagnosis of aICA is crucial because improper surgery or treatment can lead to catastrophic hemorrhage and other complications.

Patients with aICA are either asymptomatic or have nonspecific clinical symptoms [1]. According to the literature, hearing loss (48% of patients) and pulsatile tinnitus (30% of patients) are the most common symptoms of aICA [2]. Other symptoms include aural fullness, vertigo and headache [3]. The most prevalent type of hearing loss among patients with aICA is conductive hearing loss [1,4,5,6]; sensorineural and mixed hearing loss are less prevalent. Here, we discuss a case in which a patient with aICA presented with the extremely rare symptoms of mixed hearing loss and postauricular pain. Diagnosis is more difficult when a patient presents with rare symptoms. We provide detailed clinical images of the patient to illustrate how clinicians can diagnose aICA, and reviewed the literature concerning the presentation, diagnosis, and management of this disease.

## 2. Case Report

A 39-year-old female patient presented to our hospital with a 3-month history of right aural fullness and hearing impairment, and a 1-week history of right postauricular pain. She had no history of tinnitus or vertigo. She also had no history of ear trauma, ear surgery, or otitis media.

A physical examination revealed the normal and symmetric appearance of the auricles. Facial palsy was not observed. On otoscopic examination of the right ear, a red retrotympanic mass causing pulsatile protrusion of the anteroinferior quadrant of the eardrum was observed (Figure 1, Appendix A). Pure-tone audiometry (PTA) revealed mixed hearing loss in the right ear (Figure 2a).

Imaging studies were subsequently performed to assess the etiology of the mass in her tympanic cavity. Unenhanced high-resolution computed tomography (HRCT) of the temporal bone revealed lateral bulging of the right ICA to the inferomedial portion of the right tympanic cavity without a bony covering, an enlarged right inferior tympanic canaliculus, and a hypoplastic right carotid canal (Figure 3). However, no other obvious abnormalities were noted in the right external auditory canal (EAC), middle ear, inner ear, and internal auditory canal (IAC). The EAC was patent and normal in configuration. The ossicles were normal in structure. The tympanic cavity and mastoid cells were clear. The configuration of the cochlea, vestibule, and semicircular canals was normal. No obvious abnormalities were noted in the IAC, petrosal apex, and cerebellopontine angle (Figure 4). Gadolinium-enhanced magnetic resonance imaging (MRI) of the head also revealed lateral bulging in the right ICA to the right tympanic cavity (Figure 5). Serial imaging studies indicated a diagnosis of right aICA.

Because this patient’s aICA had no risk of bleeding, we decided on conservative treatment with observation and regular outpatient follow-up. The patient was initially treated with acetaminophen and diclofenac sodium for her postauricular pain, but her pain did not notably improve until she received treatment with oxcarbazepine (300 mg bid). After 10 years of clinical follow-up, the most recent PTA revealed stationary mixed hearing loss in the right ear with no other complications (Figure 2b).

## 3. Discussion

The recognition of aICA is important but challenging. In many cases, a red retrotympanic pulsatile mass can be observed during an otoscopic examination; however, in some cases, a mass is found during middle ear surgery or because of massive bleeding resulting from surgical injury to the blood vessels [5]. The most common symptoms in a patient with aICA are conductive hearing loss and pulsatile tinnitus; thus, when a patient with aICA presents with the extremely rare symptoms of mixed hearing loss and postauricular pain, diagnosis is difficult. As the number of reports of aICA increases, the predominance of the affected side and gender is not apparent. In addition, bilateral involvement is not uncommon (15% of patients) [2,6].

When a red pulsatile mass is observed in the tympanic cavity, a differential diagnosis of vascular malformations in the tympanic cavity will include glomus tumor [2], hemangioma [1], and dehiscent high jugular bulb. A differential diagnosis is performed with a combination of HRCT, MRI, and magnetic resonance angiography [1,2,7]. aICA results from embryological agenesis of the cervical segment of the ICA and collateral flow through the embryonic inferior tympanic artery. aICA indicates an enlargement of the inferior tympanic artery anastomosing with an enlargement of the caroticotympanic artery when the cervical segment of the ICA fails to develop. Thus, aICA follows the course of the inferior tympanic artery and reverses that of the caroticotympanic artery. The inferior tympanic canaliculus is, therefore, much larger than normal. The aICA, after coursing through the hypotympanum, enters the horizontal carotid canal, bypassing the absent vertical carotid canal [8]. Based on the aforementioned variation in embryological development, aICA can be identified via HRCT by the following features: an intratympanic mass, an enlarged inferior tympanic canaliculus, the absence of the vertical segment of the ICA canal, and the absence of bone covering the tympanic portion of the ICA [5]. These features were all observed in our case (Figure 3). aICA is occasionally combined with cystic cochleovestibular anomaly [3], persistent stapedial artery [9], and duplicated ICAs [10], but these anomalies were not present in our case.

In audiology, hearing loss is divided into three types: conductive, sensorineural, and mixed. According to the literature, the most prevalent type of hearing loss among patients with aICA is conductive [1,4,5,6], followed by sensorineural and mixed [7]. Mixed hearing loss is rarely reported in cases of aICA [10,11]. aICA causes conductive hearing loss and is thought to be associated with a “third mobile window” [4]. aICA complicated with a cochleovestibular anomaly can cause sensorineural hearing loss [3]. In general, mixed hearing loss means that the patient may have both middle ear and inner ear abnormalities. However, in this case, because the HRCT exhibited no obvious abnormalities in the morphology of the patient’s middle and inner ear, we were unable to determine the cause of the patient’s mixed hearing loss. Nonetheless, this report widens our understanding of the spectrum of the causes of mixed hearing loss. In this report, as the patient’s mixed hearing loss in the right ear did not affect her daily life and was stable, our treatment strategy was to follow her up regularly without further treatment for the hearing loss.

For our patient, oxcarbazepine was superior to traditional pain relievers such as acetaminophen and diclofenac sodium for relieving postauricular pain in patients with aICA. However, this is only a single clinical observation. The mechanism is unclear, and more research is needed to confirm it. Oxcarbazepine is a drug used to treat both focal and generalized seizures. It is also used alone and as an add-on therapy for people with bipolar disorder for whom other treatments have not been successful. Oxcarbazepine is a prodrug that is largely metabolized to the pharmacologically active 10-monohydroxy derivative licarbazepine (MHD). Oxcarbazepine and MHD act by blocking voltage-sensitive sodium channels, resulting in the stabilization of hyperexcited neural membranes, the inhibition of repetitive neuronal firing, and a reduction in the propagation of synaptic impulses.

In cases of aICA without bleeding complications, conservative treatment is recommended [12]. This is because iatrogenic injury during surgical treatment may lead to rapid bleeding, hemiparesis, aphasia, deafness, Horner syndrome, and intractable vertigo [2]. The reinforcement of aICA using tragal cartilage as a shield to strengthen a carotid canal defect was reported in a case with a risk of bleeding [13]. In the case of iatrogenic injury, a consensus in the literature regarding optimal management has not been reached. Once iatrogenic injury occurs, the bleeding can initially be controlled by packing the EAC and the middle ear. An endovascular procedure may then be required, followed by the possibility of an anterior cerebral stroke [1].

## 4. Conclusions

This report presents four key findings: (1) a patient with aICA may present with mixed hearing loss; (2) oxcarbazepine may improve aICA-associated postauricular pain; (3) an otolaryngologist who wishes to avoid causing iatrogenic injury including catastrophic hemorrhage must perform a thorough examination using HRCT and MRI to eliminate the potential presence of aICA; and (4) conservative treatment is recommended in cases of aICA without a risk of bleeding.

## Figures and Tables

**Figure 1 medicina-58-01672-f001:**
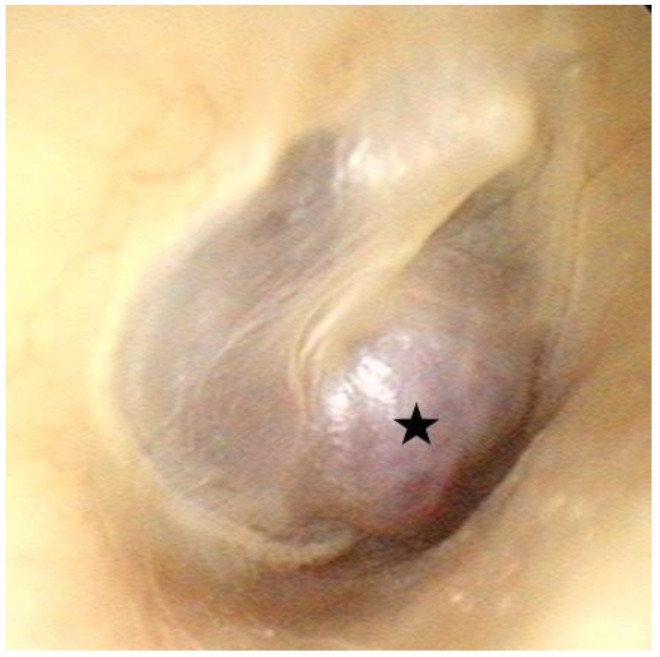
Otoscopy of right ear revealing a red retrotympanic mass (asterisk) causing pulsatile protrusion of the anteroinferior quadrant of the eardrum.

**Figure 2 medicina-58-01672-f002:**
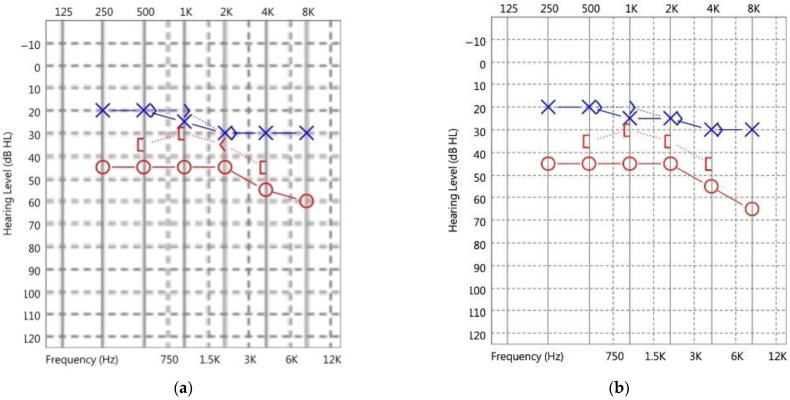
(**a**) Pure-tone audiometry (PTA) demonstrated mixed hearing loss in the right ear and mild sensorineural hearing loss at 2–8 kHz in the left ear. (**b**) After 10 years of clinical follow-up, the most recent PTA revealed stationary mixed hearing loss in the right ear. (O: unmasked air conduction of right ear, X: unmasked air conduction of left ear, 〈: unmasked bone conduction of right ear, 〉: unmasked bone conduction of left ear, [: masked bone conduction of right ear).

**Figure 3 medicina-58-01672-f003:**
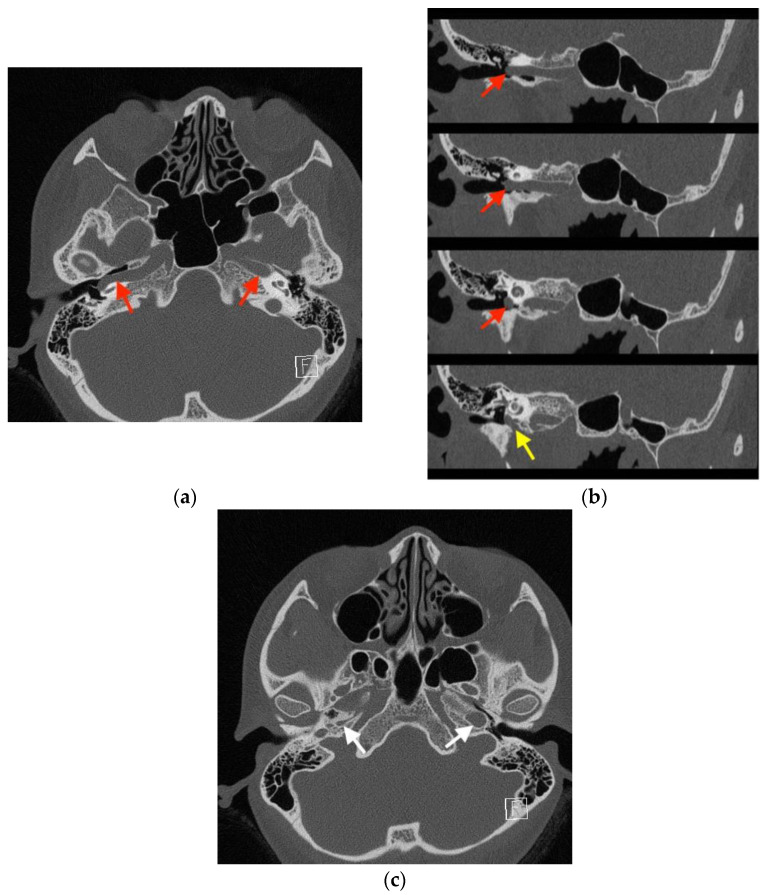
High-resolution computed tomography (HRCT) of the temporal bone, indicating a diagnosis of right aberrant internal carotid artery in the middle ear (aICA). (**a**) In the axial view, HRCT revealed a normal left internal carotid artery (ICA) (red arrow on the patient’s left) and lateral bulging of the right ICA to the right of the tympanic cavity without a bony covering (red arrow on the patient’s right). (**b**) The oblique view demonstrated the lateral bulging of the right ICA to the right of the tympanic cavity (red arrows) and the enlarged right inferior tympanic canaliculus (yellow arrow) through which the aICA (inferior tympanic artery portion) passes. (**c**) In the axial view, a normal left carotid canal (white arrow on the patient’s left) and a hypoplastic carotid canal (white arrow on the patient’s right) were observed. The aforementioned findings from HRCT indicated a diagnosis of right aICA.

**Figure 4 medicina-58-01672-f004:**
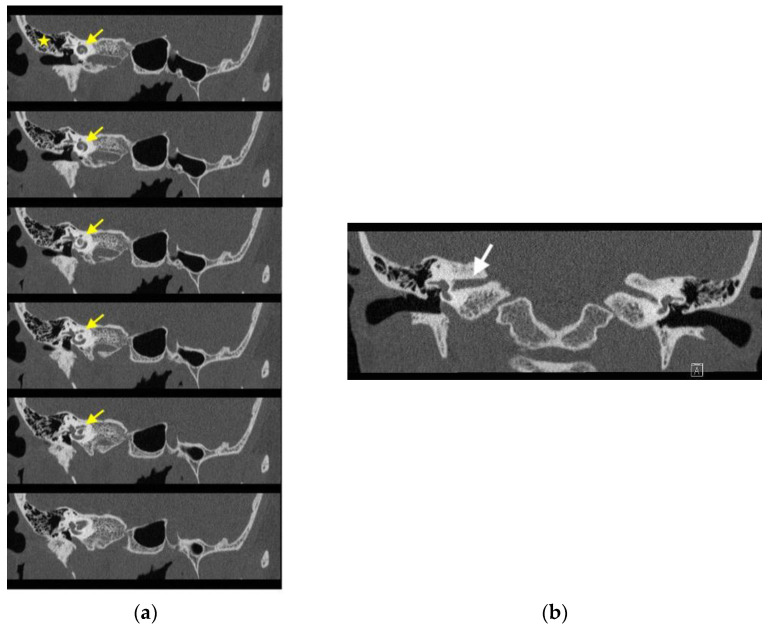
High-resolution computed tomography (HRCT) revealing no significant abnormalities in the right external auditory canal (EAC), middle ear, inner ear, and internal auditory canal (IAC). (**a**) In the oblique view, the EAC was patent and normal in configuration. The tympanic cavity and mastoid cells (asterisk) were clear. The configuration of the cochlea (yellow arrows), vestibule, and semicircular canals was normal. (**b**) In the coronal view, no significant abnormalities were noted in the IAC (white arrow) and petrosal apex.

**Figure 5 medicina-58-01672-f005:**
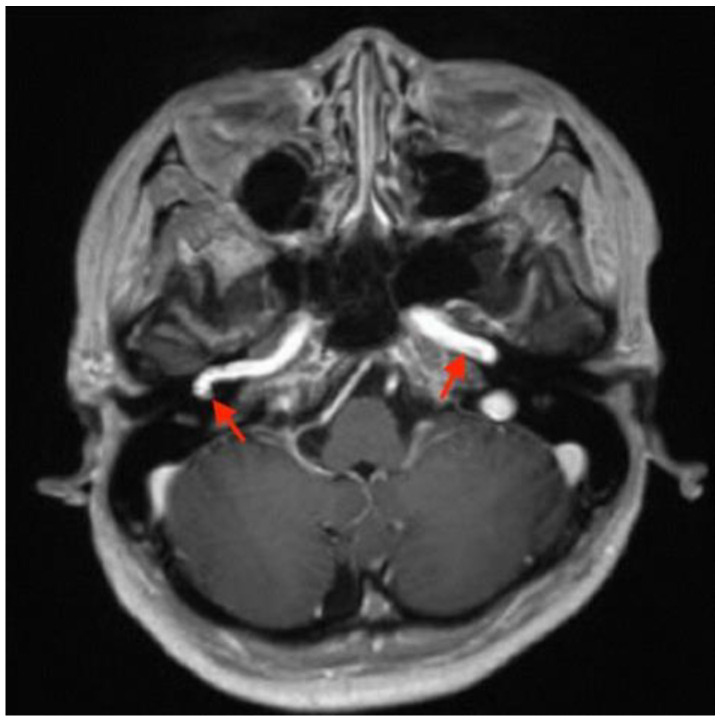
Gadolinium-enhanced magnetic resonance image revealing a normal left internal carotid artery (ICA) (arrow on the patient’s left) and lateral bulging of the right ICA to the right of the tympanic cavity (arrow on the patient’s right) in the axial view.

## Data Availability

Not applicable.

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
