# Peer review of "Patient with Aberrant Internal Carotid Artery in the Middle Ear Presenting with Rare Symptoms of Mixed Hearing Loss and Postauricular Pain: A Case Report"

_medicina, 2022, doi:10.3390/medicina58111672_

Round 1

Reviewer 1 Report

I would like to thank the authors for contributing to the medical literature by reporting this case. It is one-of-a-kind and well-presented, with great figures and explanations. 

I recommend its publication in its current form.

Author Response

I am very thankful for the comments and suggestions from the reviewer. The reviewer's approval of this case report will encourage me to work harder on submitting papers in the future.

Reviewer 2 Report

1.The patient was reported to have mixed hearing loss and postauricular pain. Oxcarbazepine was given for the treatment of postauricular pain, but the treatment of hearing loss was not mentioned. Please explain whether it was not treated because it did not affect daily life.

2. The treatment of postauricular pain is only for symptoms, and the effect of medication is not universal. Conclusion and discussion section of the article should not focus on the efficacy of oxcarbazepine monotherapy.

3. There are many cases reported in the previous literature, and the basic content of anatomical variation can be increased.

4. No meaningful diagnosis and treatment experience was obtained.

Author Response

Point 1: The patient was reported to have mixed hearing loss and postauricular pain. Oxcarbazepine was given for the treatment of postauricular pain, but the treatment of hearing loss was not mentioned. Please explain whether it was not treated because it did not affect daily life.

Response 1: I have followed the reviewer's suggestion and added the explanation at the end of the third paragraph of the discussion section. → "In this report, as the patient's mixed hearing loss in the right ear did not affect her daily life and was stable, our treatment strategy was to follow up regularly without further treatment for the hearing loss."

Point 2: The treatment of postauricular pain is only for symptoms, and the effect of medication is not universal. Conclusion and discussion section of the article should not focus on the efficacy of oxcarbazepine monotherapy.

Response 2: I have followed the reviewer's suggestion and added the explanation in the fourth paragraph of the discussion section. → "However, this is only a single clinical observation. The mechanism is unclear, and more research is needed to confirm it."

In the conclusion section, regarding the efficacy of oxcarbazepine, I have modified it to a more conservative word.

Point 3: There are many cases reported in the previous literature, and the basic content of anatomical variation can be increased.

Response 3: I have followed the reviewer's suggestion and increased the basic content of anatomical variation in the second paragraph of the discussion section. → "aICA indicates an enlargement of the inferior tympanic artery anastomosing with an enlargement of the caroticotympanic artery when the cervical segment of the ICA fails to develop. Thus, aICA follows the course of the inferior tympanic artery and reverses that of the caroticotympanic artery. The inferior tympanic canaliculus is, therefore, much larger than normal. aICA, after coursing through the hypotympanum, enters the horizontal carotid canal bypassing the absent vertical carotid canal [8]."

Point 4: No meaningful diagnosis and treatment experience was obtained.

Response 4: In this report, we provide detailed clinical images (including rare video recordings) of patient to illustrate how clinicians diagnose aICA. After ten years of long-term follow-up, this report supports conservative management of aICA without bleeding complications. We also remind readers that special attention is required for iatrogenic injury of aICA. We believe this report will be very helpful to readers.

Reviewer 3 Report

Comment

The author reported the case with aberrant internal carotid artery in the middle ear. The case was rare malformation and presented with mixed-hearing loss and postauricular pain. The case of this paper is interesting, I think this manuscript will be acceptable for publication after minor revision.

Minor revision

The author should present the results of the pure tone audiogram and HRCT both ten years ago and now.

Author Response

Point 1: The author should present the results of the pure tone audiogram and HRCT both ten years ago and now.

Response 1: I have added the most recent pure-tone audiogram in the revised manuscript according to the reviewer's suggestion. The most recent pure-tone audiogram is marked as figure 2(b) in the revised manuscript.
In my treatment strategy, if the patient's audiogram and clinical symptoms change, I will arrange HRCT. However, so far, since the patient's audiogram and symptoms are stable, I have not arranged HRCT for follow-up.

Reviewer 4 Report

This is a well written case report with excellent illustrations and solid documentation.   It was also well researched.    The authors have a unique treatment strategy for this patient and for that reason should have some interest to readers but it is still an anecdotal experience because of its rarity .   It is still a fascinating case and should remind clinicians of the inherent dangers when visualizing a reddish mass in the middle ear.  This particular vascular anomaly has disastrous  consequences if the integrity of the artery is violated and the paper reinforces that potential risk.   

Author Response

(The authors gave the same response as above.)

Round 2

Reviewer 2 Report

The article as a whole has no special problems and agrees to be published in its current form.

Author Response

Reviewer's comments: The article as a whole has no special problems and agrees to be published in its current form.

Response: I am very thankful for the comments and suggestions from the reviewer. The reviewer's approval of this case report will encourage me to work harder on submitting papers in the future.